# SPIDER: Searching Personalized Neural Architecture for Federated Learning

## Abstract

Federated learning (FL) is an efficient learning framework that assists distributed machine learning when data cannot be shared with a centralized server. Recent advancements in FL use predefined architecture-based learning for all clients. However, given that clients' data are invisible to the server and data distributions are non-identical across clients, a predefined architecture discovered in a centralized setting may not be an optimal solution for all the clients in FL. Motivated by this challenge, we introduce SPIDER, an algorithmic framework that aims to **S**earch **P**ersonal**I**zed neural architecture for fe**DER**ated learning. SPIDER is designed based on two unique features: (1) alternately optimizing one architecture-homogeneous global model (Supernet) in a generic FL manner and one architecture-heterogeneous local model that is connected to the global model by weight-sharing-based regularization (2) achieving architecture-heterogeneous local model by an operation-level perturbation based neural architecture search method. Experimental results demonstrate that SPIDER outperforms other state-of-the-art personalization methods with much fewer times of hyperparameter tuning.

## 1 Introduction

Federated Learning (FL) is a promising decentralized machine learning framework that facilitates data privacy and low communication costs. It has been extensively explored in various machine learning domains such as computer vision, natural language processing, and data mining. Despite many benefits of FL, one major challenge involved in FL is data heterogeneity, meaning that the data distributions across clients are not identically or independently (non-I.I.D) distributed. The non-I.I.D distributions result in the varying performance of a globally learned model across different clients. In addition to data heterogeneity, data invisibility is another challenge in FL. Since clients' private data remain invisible to the server, from the server's perspective, it is unclear how to select a pre-defined architecture from a pool of all available candidates. In practice, it may require extensive experiments and hyper-parameter tuning over different architectures, a procedure that can be prohibitively expensive.

To address the data-heterogeneity challenge, variants of the standard FedAvg have been proposed to train a global model, including the `FedProx` Li et al. (2018), `FedOPT` Reddi et al. (2020), and `FedNova` Wang et al. (2020). In addition to training of a global model, frameworks that focus on training personalized models have also gained a lot of popularity. The `Ditto` Li et al. (2021b), `PerFedAvg` Fallah et al. (2020a), and `pFedMe` Dinh et al. (2020) are some of the recent works that have shown promising results to obtain improved performance across clients. However, all these works exploit pre-defined architectures and operate at the optimization layer. Consequently, in addition to their inherent hyper-parameter tuning, these personalization frameworks often encounter the data-invisibility challenge that one has to select a suitable model architecture involving a lot of hyper-parameter tuning.

In this work, we adopt a different and complementary technique to address the data heterogeneity challenge for FL. We introduce SPIDER, an algorithmic framework that aims to **S**earch **P**ersonal**I**zed neural architecture for fe**DER**ated learning. Recall that in a centralized setting, the neural architecture search (NAS) aims to search for optimal architecture to address system design challenges such as lower latency Wu et al. (2019), lesser memory cost Li et al. (2021a), and smaller energy consump-

tion Yang et al. (2020). For architecture search, there are three well-known methods explored in literature, gradient-based Liu et al. (2018), evolutionary search Liu et al. (2021), and reinforcement learning Jaafra et al. (2019). Out of these, gradient-based methods are generally considered more efficient because of their ability to yield higher performance in comparatively lesser time Santra et al. (2021).

To achieve personalization at the architecture level in FL, we propose a unified framework, SPIDER. This framework essentially deploys two models, local and global models, on each client. Initially, both models use the DARTS search space-based Supernet Liu et al. (2018), an over-parameterized architecture. In the proposed framework, the global model is shared with the server for the FL updates and, therefore, stays the same in the architecture design. On the other hand, the local model stays completely private and performs personalized architecture search, therefore, gets updated. To search for the personalized child model, SPIDER deploys SPIDER-Searcher on each client's local model. The SPIDER-Searcher is built upon a well-known gradient-based NAS method, named perturbation-based NAS Wang et al. (2021). The main objective of the SPIDER framework is to allow each client to search and optimize their local models while benefiting from the global model. To achieve this goal, we propose an alternating bi-level optimization-based SPIDER Trainer that trains local and global models in an alternate fashion. However, the main challenge here is the optimization of an evolving local model architecture while benefiting from a fixed global architecture. To address this challenge, SPIDER Trainer performs weight-sharing-based regularization that regularizes the common connections between the global model's Supernet and the local model's child model. This aids clients in searching and training heterogeneous architectures tailored for their local data distributions. In a nutshell, this approach not only yields architecture personalization in FL but also facilitates model privacy (in the sense that the derived child local model is not shared with the server at all).

To evaluate the performance of the proposed algorithm, we consider a cross-silo FL setting and use Dirichlet distribution to create non-I.I.D data distribution across clients. For evaluation, we report test accuracy at each client on the 20% of training data kept as test data for each client. We show that the architecture personalization yields better results than state-of-the-art personalization algorithms based solely on the optimization layer, such as Ditto Li et al. (2021b), perFedAvg Fallah et al. (2020a), and local adaptation Cheng et al. (2021).

To summarize, the following are the key contributions of our work.

• We propose and formulate a personalized neural architecture search framework for FL named SPIDER, from a perspective complementary to the state-of-the-art to address data heterogeneity challenges in FL.

• SPIDER is designed based on two unique features: (1) maintaining two models at each client, one to communicate with the server and the other to perform a local progressive search, and (2) operating local search and training at each client by an alternating bilevel optimization and weight sharing-based regularization along the FL updates.

• We run extensive experiments to demonstrate the benefit of SPIDER compared with state-of-the-art personalized FL approaches such as Ditto Li et al. (2021b), perFedAvg Fallah et al. (2020a) and Local Adaptation Cheng et al. (2021) on three datasets, CIFAR10, CIFAR100 and CINIC10. With the ResNet18 model, on the CIFAR10 dataset with heterogeneous distribution, we demonstrate an increase of the average local accuracy by 2.8%, 1.7%, and 5.5%, over Ditto, PerFedAvg, and Local Adaption, respectively. Further, on CIFAR100, we demonstrate an accuracy gain of 10%, 6%, 4% over Ditto, Local Adaptation, and perFedAvg, respectively. Likewise, on CINIC-10, we demonstrate an accuracy gain of 20%, 23%, and 24% over Ditto, Local Adaptation, and perFedAvg, respectively.

## 2 RELATED WORKS

**Heterogeneous Neural Architecture for FL** Heterogeneous neural architecture is one way to personalize the model in FL. For personalization, the primal-dual framework Smith et al. (2017a), clustering Sattler et al. (2020), fine-tuning with transfer learning Yu et al. (2020b), meta-learning Fallah et al. (2020a), regularization-based method Hanzely & Richtárik (2020); Li et al. (2021b) are among the popular methods explored in the FL literature. Although these techniques achieve improved personalized performance, all of them use a pre-defined architecture for each client. Het-

eroFL Diao et al. (2020) is a recent work that accomplishes the aggregation of heterogeneous models by assigning sub-parts of the global model based on their computation budget and aggregating the parameters common between different clients. Similarly, work Luo et al. performs a limited channel-wise search to assign sub-models meeting clients' efficiency budgets and performs the partial aggregation of weights at the server. Another work Lin et al. (2020b) accomplishes this task by forming clusters of clients of the same model and allowing for heterogeneous models across clusters. Model aggregation is based on cluster-wise aggregation followed by a knowledge distillation from the aggregated models into the global model. Given data invisibility in FL, deciding which architecture would work for which client is a challenging task and requires exploration. Another work Dudziak et al. (2022) personalizes architectures to compute-power-based clusters instead of individual clients. As such, our proposed method aims to achieve personalized architecture automatically and focuses on the objective of tailoring architecture to individual clients' data distribution.

**Neural Architecture Search for FL**    Neural Architecture Search (NAS) has gained momentum in recent literature to search for a global model in a federated setting. FedNAS He et al. (2020b) explores the compatibility of MileNAS solver with Fed averaging algorithm to search for a global model. Direct Federated NAS Hu et al. (2020) is another work in this direction that explores the compatibility of a one-shot NAS method, DSNAS Hu et al. (2020), with Fed averaging algorithm with the same application, in search of a global model. Zhu & Jin (2021) uses evolutionary NAS to design a master (global) model. Singh et al. (2020) explores the concept of differential privacy using DARTs solver Liu et al. (2018) to explore the trade-off between the accuracy and privacy of a global model. Xu et al. (2020) starts with a pre-trained handcraft model and continues pruning the model until it satisfies the efficiency budget. Another work Hoang & Kingsford (2021) divides the model architecture into global and personal components, and searches the personal component for personalization on identical and independent (IID) vision tasks. Where all these models search for a unified global model or personalized components of a shared model, a key distinction of our work with these works is that we aim to search for a complete personalized model for each client.

## 3    PRELIMINARIES, MOTIVATION, AND DESIGN GOALS

In this section, we introduce the state-of-the-art methods for personalized federated learning, discuss the motivation for personalizing model architectures, and summarize our design goals.

**Personalized Federated Learning**    A natural formulation of FL is to assume that among $c$ distinct clients, each client $k$ has its own distribution $P_i$, draws data samples from $P_i$, and aims to solve a supervised learning task (e.g., image classification) by optimizing a global model $\boldsymbol{w}$ with other clients collaboratively. At a high-level abstraction, the optimization objective is then defined as:

$$\min_{\boldsymbol{w}^*} G\left(F_1(\boldsymbol{w}), ..., F_c(\boldsymbol{w})\right), \tag{1}$$

where $F_k(\boldsymbol{w})$ measures the performance of the model global $\boldsymbol{w}$ on the private dataset at client $k$ (local objective), and $G$ is the global model aggregation function that aggregates each client's local objectives. For example, for FedAvg, $G(.)$ would be weighted aggregation of the local objectives, $\sum_{k=1}^{c} p_k F_k(\boldsymbol{w})$, where $\sum_{k=1}^{c} p_k = 1$.

However, as distributions across individual clients are typically heterogeneous (i.e., non-I.I.D.), there is a growing line of research that advocates reformulating FL as a personalization framework, dubbed as personalized FL (PFL). In PFL, the objective is redirected to find a personalized model $\boldsymbol{v}_k$ for device $k$ that performs well on the local data distribution:

$$\min_{\boldsymbol{v}_1^*, ..., \boldsymbol{v}_c^*} \left(F_1(\boldsymbol{v}_1), ..., F_c(\boldsymbol{v}_c)\right), \tag{2}$$

To solve this challenging problem, various PFL methods are proposed, including FedAvg with local adaptation (Local-FL) Cheng et al. (2021); Yu et al. (2020a); Wang et al. (2019), MAML-based PFL (MAML-FL) Fallah et al. (2020b); Jiang et al. (2019), clustered FL (CFL) Ghosh et al. (2020); Sattler et al. (2021), personalized layer-based FL (PL-FL) Liang et al. (2020); Hoang & Kingsford (2021); Pillutla et al. (2022); Hoang & Kingsford (2021), federated multitask learning (FMTL) Smith et al. (2017b) and knowledge distillation (KD) Lin et al. (2020a); He et al. (2020a).

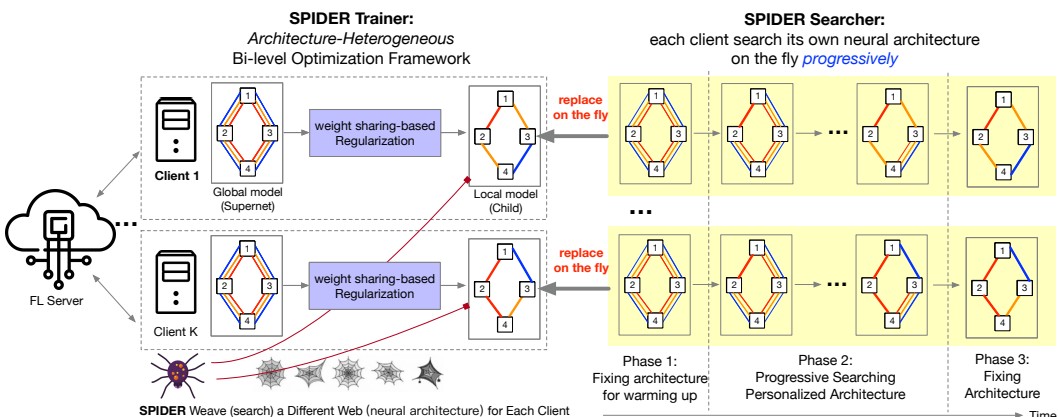

*Figure 1: Illustration of SPIDER framework. SPIDER is weaving (searching) a different web (neural architecture) for each client.*

**Motivation for Neural Architecture Personalization**   Distinct from these existing works on PFL, we propose a new approach to instead personalize model architecture for each client. We are motivated by one critical potential benefit, that is, the searched architecture for each client is expected to fit its own distinct distribution, which has the potential to provide a substantial improvement over the existing PFL baselines that only personalize model weights. In addition, a personalized architecture search allows the clients to even keep their local model architectures private in a sense the server and other clients neither know the architecture nor the weights of that architecture. This further enhances the privacy guarantees of FL and is helpful in business cases that each client hopes to also protect its model architecture.

**Design Goals**   Our goal is to enable personalized neural architecture search for all clients in FL. In this context, the limitation of existing personalized FL methods is obvious: Local-FL and MAML-FL need every client to have the same architecture to perform local adaptations; In CFL, the clustering step requires all clients to share a homogeneous model architecture; PL-FL can only obtain heterogeneous architectures for a small portion of personalized layers, but it does not provide an architecture-agnostic method to determine the boundary of personalized layers in an automated mechanism; FMTL is a regularization-based method which cannot perform regularization when architectures are heterogeneous across clients; KD has an unrealistic assumption that the server has enough public dataset as the auxiliary data for knowledge distillation.

To circumvent these limitations, our goal is to design an architecture-personalized FL framework with the following requirements:

- **R1**: *allowing heterogeneous architectures for all clients*, which can capture fine-grained data heterogeneity;
- **R2**: *searching and personalizing the entire architecture space*, to avoid the heuristic search for the boundary of personalized layers;
- **R3**: *requiring no auxiliary data at the client- or server-side (unlike knowledge distillation-based PFL)*;

We now introduce SPIDER which meets the above requirements in a unified framework.

## 4 METHODOLOGY: SPIDER

### 4.1 OVERVIEW

The overall framework of SPIDER is illustrated in Figure 1. Essentially, each client maintains two models in this framework: one architecture-homogeneous global model for collaborative training with other clients, and one architecture-heterogeneous local model that initially shares the same super architecture space as the global model. At a high-level, SPIDER is formulated as an **architecture-personalized bi-level optimization** problem (Section 4.2) and proposes the solver as

the orchestration of **SPIDER Trainer** (Section 4.3) and **SPIDER-Searcher** (Section 4.4). **SPIDER Trainer** is an architecture-personalized training framework that can collaboratively train heterogeneous neural architectures across clients. To allow federated training on the expected heterogeneous local architectures, it enables regularization between an arbitrary personalized architecture and the global model via weight sharing. With this support, **SPIDER-Searcher** is designed to *dynamically adjust the architecture of each client's local model on the way*. To search a personalized architecture for the local data distribution of each client, SPIDER-Searcher is built on a novel neural architecture search (NAS) method that searches optimal local Subnet progressively using operation-level perturbation on the accuracy value as the criterion. Overall, each client's local model goes through three phases (also shown in Figure 1): pre-training to warm up the initial local model, progressive neural architecture search, and final training of the searched architecture-personalized model.

SPIDER can meet the design goals **R1**-**R3** introduced in Section 3 because 1) each client performs independent architecture personalization with its own private data (**R1**), 2) the search space is not restricted to a portion of the model (**R2**), and 3) no auxiliary data is used to assist the search and train process (**R3**).

## 4.2 SPIDER FORMULATION: ARCHITECTURE-PERSONALIZED BI-LEVEL OPTIMIZATION

SPIDER aims to personalize (weave) a different neural architecture (web) for each client. To generate heterogeneous architectures across clients, we use two models, a local model $(\mathcal{A}_k)$ and a global model (Supernet $\mathcal{A}$) at each client, and formulate SPIDER as an architecture-personalized bi-level optimization problem for each client $k \in \{1, 2, ..., c\}$:

$$\min_{\boldsymbol{v}_k, \mathcal{A}_k \subseteq \mathcal{A}} \quad F_k(\boldsymbol{v}_k, \mathcal{A}_k; \boldsymbol{w}^*, \mathcal{A}) \tag{3}$$

$$\text{s.t.} \quad \boldsymbol{w}^* \in \arg\min_{\boldsymbol{w}} G\left(F_1(\boldsymbol{w}, \mathcal{A}), ..., F_K(\boldsymbol{w}, \mathcal{A})\right), \tag{4}$$

where $F_k$ is the local objective of the client $k$; $\boldsymbol{w}$, and $\boldsymbol{v}_k$ are all learnable parameters; $\boldsymbol{w}$ denotes the parameter of the global model architecture $\mathcal{A}$, $\boldsymbol{v}_k$ is the weight parameter of the local personalized architecture $\mathcal{A}_k$ of the client $k$. Here, $\mathcal{A}_k$ is a child neural architecture of a Supernet $\mathcal{A}$ such that $\mathcal{A}_k \subseteq \mathcal{A}$. The Supernet $\mathcal{A}$ can be considered having a mask $M$, each entry representing a learnable parameter $\alpha_{ij}$, $i \in \{1, 2, .., e\}$ and $j \in \{1, 2, ..., o\}$ where $e$ is the total number of edges and $o$ is the total number of operations at each edge. Hence, the size of mask $M$ is equal to $e \times o$. For the Supernet mask $M$, all $\alpha_{ij}$ entries are 1. Likewise, we maintain a mask $M_k$ for local models $\mathcal{A}_k$. Our goal is to optimize $\alpha_{ij}$ parameters of the mask $M_k$ such that we learn child architecture $\mathcal{A}_k = M_k \odot \mathcal{A}$. Note that in Eq.(4), we aim to learn a global model $\mathcal{A}$ in a federated learning setting, which formulates our outer optimization. However, in the inner optimization given in Eq.(3), the objective of each client is to optimize its local model's architecture $\mathcal{A}_k$ and its associated parameters $\boldsymbol{v}_k$ while benefiting from the global model $\boldsymbol{w}^*$. As a tractable, yet general case study, SPIDER reuses the DARTS architecture space as Supernet $\mathcal{A}$ that contains a set of edges $\{1, ..., e\}$, and each edge has multiple operations $\{1, ..., o\}$. Based on this definition, $\mathcal{A}_k$ maintains the operation-level granularity: $\mathcal{A}_k$'s edge set space is a subset of $\mathcal{A}$'s edge set space, and the operation set in $\mathcal{A}_k$'s each edge may also be a subset space. We provide further details of the search space in Appendix A.3.

**The difficulty of jointly optimizing the architecture** $\mathcal{A}_k$ **and related weight parameters** $v_k$     The key difference of our formulation from existing bi-level optimization for FL (e.g., Li et al. (2021b)) is that in our case, $\mathcal{A}_k$ is also learnable (Eq.(3)). We assume each client can have an evolving architecture $\mathcal{A}_k$, i.e., Eq.(3) has to optimize the architecture $\mathcal{A}_k$ and its related weight parameters $\boldsymbol{v}_k$ jointly, while using complete Supernet-based global model weights, $\boldsymbol{w}$. SPIDER addresses this challenge by the orchestration of SPIDER-Trainer and SPIDER-Searcher.

## 4.3 SPIDER TRAINER: FEDERATED TRAINING ON HETEROGENEOUS ARCHITECTURES

In this section, we describe SPIDER trainer, an architecture-personalized training framework that can collaboratively train heterogeneous neural architectures across clients.

To clearly show how SPIDER handles the optimization difficulty of Eq.(3), we first downgrade the objective to the case that all clients use *predefined* (fixed) heterogeneous architectures (derived from

the Supernet $\mathcal{A}$). More specially, we reduce the aforementioned optimization framework in Eq.(3) and Eq.(4) to the following:

$$\min_{\boldsymbol{v}_k} \quad h_k(\boldsymbol{v}_k, \mathcal{A}_k; \boldsymbol{w}^*, \mathcal{A}) = F_k(\boldsymbol{v}_k) + \frac{\lambda}{2}||\boldsymbol{v}_k - \boldsymbol{w}^*_{share}||^2 \tag{5}$$

$$\text{s.t.} \quad \boldsymbol{w}^* \in \arg\min_{\boldsymbol{w}} G\left(F_1(\boldsymbol{w}, \mathcal{A}), ..., F_K(\boldsymbol{w}, \mathcal{A})\right), \tag{6}$$

where local model's weights $\boldsymbol{v}_k$ are regularized towards the global model $\boldsymbol{w}^*_{share}$, where $\boldsymbol{w}^*_{share}$ are the weight parameters of the operation set space of $\mathcal{A}$ overlapping (sharing) with $\mathcal{A}_k$. Also, $\lambda$ is the regularization hyper-parameter. Note that, now, only $\boldsymbol{v}_k$ needs to be optimized in Eq.5, while $\mathcal{A}_k$ is fixed during the optimization.

We then solve Eq.5 and Eq.6 alternately. We summarize this optimization procedure as SPIDER-Trainer with a detailed pseudo code illustrated in Algorithm 1. In this algorithm, we can note that the global model (line #12) and the local model (line #15) are updated alternately. The strength of this algorithm lies in its elaborate design, which provides the following key benefits:

---

**Algorithm 1 SPIDER Trainer**

---

1: **Initialization:** initialize $c$ clients with the $k$-th client has a global model $\boldsymbol{w}_k$ using Supernet $\mathcal{A}$, and a local model $\boldsymbol{v}_k$ using subnet $\mathcal{A}_k$ with mask $M_k$(set $\mathcal{A}_k = \mathcal{A}$ at the begining); $p$ is the number of local epochs; $r$ is the number of rounds; $t_s$ number of rounds to start search; $\tau$ is the recovery periods in the units of rounds.
2: **Server executes:**
3:     **for** each round $t = 0, 1, 2, ..., r - 1$ **do**
4:         **for** each client $k$ **in parallel do**
5:             $\boldsymbol{w}_k^{t+1} \leftarrow \text{ClientLocalSearch}(k, \boldsymbol{w}^t, t)$
6:         $\boldsymbol{w}^{t+1} \leftarrow \sum_{k=1}^{K} \frac{N_k}{N} \boldsymbol{w}_k^{t+1}$
7:
8:     *function* **ClientLocalSearch**$(k, w^t, t)$: *// Run on client $k$*
9:     **for** each epoch in $p$ **do**
10:         **for** minibatch in training and validation data **do**
11:             $\mathcal{A}_k^{t+1}, M_k^{t+1} = \textbf{ProgressiveNAS}(\mathcal{A}_k^t, M_k^t, t_s, \tau, t)$
12:             Update Global model: $\boldsymbol{w}^{t+1} = \boldsymbol{w}^t - \eta_w \nabla_w \mathcal{L}_k^{\text{tr}}(\boldsymbol{w}^t)$
13:             **for** each $\alpha_{ij}$ in $M_k^{t+1}$ **do**
14:                 $w_{share\,ij}^{t+1} = w_{ij}^{t+1} \odot \alpha_{ij}$ *// overlapping operation set space between $\mathcal{A}$ and $\mathcal{A}_k$*
15:             Update Local Model: $\boldsymbol{v}_k^{t+1} = \boldsymbol{v}_k^t - \eta_v \left(\nabla_v \mathcal{L}_k^{\text{tr}}(\boldsymbol{v}_k^t) + \lambda(\boldsymbol{v}_k^t - \boldsymbol{w}_{share}^{t+1})\right)$
16:     **return** $w$ to server

---

**(1) Enabling regularization between an arbitrary personalized architecture and the global model** Most importantly, SPIDER-Trainer connects each personalized model with the global model by enabling the regularization between two different architectures: an arbitrary personalized architecture for the local model $\mathcal{A}_k$ of client $k$ and the global model with Supernet $\mathcal{A}$. This is done by weight sharing. $\boldsymbol{w}_{share}^*$ is essentially used to regularize a subnet ($\mathcal{A}_k$) model parameters $\boldsymbol{v}_k$ towards the global model shared/common parameters $\boldsymbol{w}_{share}^*$, as shown in Eq. 6.

**(2) Avoiding heterogeneous aggregation** SPIDER-Trainer avoids the aggregation of heterogeneous model architectures at the server side. As such, no sophisticated and unstable aggregation methods are required (e.g. knowledge distillation Lin et al. (2020a), etc.), and it is flexible to use other aggregation methods beyond FedAvg (e.g., Karimireddy et al. (2020); Reddi et al. (2021)) to update the global model.

**(3) Enabling architecture privacy** In this algorithm, only the global model is transmitted between the client and the server. This enables architecture privacy because each client's architecture is hidden from the server and other clients.

**(4) Potential robustness to adversarial attacks** The weight sharing-based regularization not only yields the benefit of personalization in FL but also makes the FL framework more robust to adversarial attacks. Its robustness advantage comes from its ability to keep the local model private and regularize towards the global model based on its regularization parameter, as shown before by architecture-homogeneous bi-level optimization Li et al. (2021b).

### 4.4 SPIDER-SEARCHER: PERSONALIZING ARCHITECTURE

Although SPIDER trainer is able to collaboratively train heterogeneous architectures, manual design of the architecture for each client is impractical or suboptimal. As such, we further add a neural architecture search (NAS) component, SPIDER-Searcher, in Algorithm 1 (line #11) to adapt $\mathcal{A}_k$ to its local data distribution in a progressive manner. We now present the details of the SPIDER-Searcher.

**Progressive Neural Architecture Search** Essentially, SPIDER-Searcher dynamically changes the architecture of $\mathcal{A}_k$ during the entire federated training process. This is feasible because the weight sharing-based regularization can handle an arbitrary personalized architecture (introduced in Section 4.3). Due to this characteristic, SPIDER-Searcher can search $\mathcal{A}_k$ in a progressive manner (shown in Figure 1): **Phase 1**: At the beginning, $\mathcal{A}_k$ is set equal to Supernet $\mathcal{A}$. The intention of SPIDER-Searcher in this phase is to warm up the training of the initial $\mathcal{A}_k$ so it does not change $\mathcal{A}_k$ for a few rounds; **Phase 2**: After warming up, SPIDER-Searcher performs edge-by-edge search gradually. In each edge search, only the operation with the highest impact on the accuracy is kept. It also uses a few rounds of training as a recovery time before proceeding to the next round of search. This process continues until all edges finish searching; **Phase 3**: After all edges finish searching, SPIDER-Searcher does not change $\mathcal{A}_k$. This serves as a final training of the searched architecture-personalized model. This three-phase procedure is summarized as Algorithm 2. Now, we proceed to elaborate on how we calculate the impact of an operation on the Supernet.

---

**Algorithm 2 SPIDER-Searcher**

---

1: **Search Space:** in the architecture $\mathcal{A}_k^t \subseteq \mathcal{A}$, $\mathcal{E}$ is the super set of all edges $\{1, ..., e\}$, $\mathcal{E}_s$ is the remaining subset of edges that have not been searched, and each edge $e$ has multiple operations $\{1, ..., o\}$.
2: *function* **ProgressiveNAS**($\mathcal{A}_k^t, M_k^t, t_s, \tau$, t)
3:    **if** $t \geq t_s$ and $t \% \tau == 0$ and LEN($\mathcal{E}_s$) $> 0$ **then**
4:       $i$ = RANDOM ($\mathcal{E}_s$) // random selection
5:       // searching without training
6:       **for all** operation $j$ on edge $i$ **do**
7:          evaluate validation accuracy of $\mathcal{A}_k$ when operation $\alpha_{ij}$ is set to zero (removed)($ACC_{\backslash \alpha_{ij}}$)
8:       in the $ith$ row of $M_k^t$, keep only one operation $\alpha_{ij} = 1$ corresponding to the lowest value of ($ACC_{\backslash \alpha_{ij}}$).
9:       remove $i$ from $\mathcal{E}_s$, update $\mathcal{A}_k^{t+1} = \mathcal{A}_k^t \odot M_k^t$.
10:    **else**
11:       return $\mathcal{A}_k^t$ and $M_k^t$ directlly
12:    **return** updated $\mathcal{A}_k^{t+1}$ and $M_k$ after selection

---

**Operation-level perturbation-based selection** In phase 2, we specify selecting the operation with the highest impact using operation-level perturbation. More specially, instead of optimizing the mixed operation architecture parameters $\alpha$ using another bi-level optimization as DARTS (a.k.a. gradient-based search) to pick optimal operation according to magnitude of $\alpha$ parameters (*magnitude-based selection*), we assign a constant value to $\alpha$ and use *the impact of an operation on the local validation accuracy* (perturbation) as a criterion to search on the edge. This simplified method is much more efficient given that it only requires evaluation-based search rather than training-based search (optimizing $\alpha_{ij}$). In addition, it avoids inserting another bi-level optimization for NAS inside a bi-level optimization for FL, making the framework stable and easy to tune. Finally, this method avoids suboptimal architecture Wang et al. (2021) lead by magnitude-based selection in differentiable NAS.

## 5 EXPERIMENTS

This section presents the experimental results of the proposed method, SPIDER. All our experiments are based on non-IID data distribution among FL clients. We have used latent Dirichlet Distribution (LDA), which is a common data distribution in FL to generate non-IID data across clients He et al. (2020c), Yurochkin et al. (2019).

## 5.1 EXPERIMENTAL SETUP

**Task and Dataset**   We perform an image classification task on three well-known datasets, **CIFAR10**, **CIFAR100** and **CINIC10**. CIFAR10 dataset consists of 60000 32x32 color images in 10 classes, with 6000 images per class, and CIFAR100 dataset consists of 60,000 images in 100 classes, with 600 images per class. CIFAR100 has more classes and comparatively fewer data per class, therefore, it is considered more challenging than CIFAR10. In addition, CINIC10 consists of 270,000 32x32 color images in 10 classes, with 90,000 images per train, test, and validation subset. CINIC10 dataset is a larger dataset and includes images from ImageNet as well. We generate non-IID data across 8 clients by exploiting LDA distribution with parameter ($\alpha = 0.2$) for the training data of CIFAR10, CIFAR100, and CINIC10 datasets. The LDA distribution of CIFAR10 and CIFAR100 has been shown in Appendix A.1.

For personalized architecture experiments with SPIDER, we split the total training data samples present at each client into training (50%), validation (30%), and testing sets (20%). For other personalization schemes used for comparison, we do not need validation data. Therefore, we split the data samples of each client with training (80%) and test (20%) for a fair comparison. In addition, we fix the non-IID data distribution in all experiments. We provide Hyperparameter search details in Appendix A.2.

Table 1: *Average local validation Accuracy Comparison of SPIDER with other personalization techniques on CIFAR10, CIFAR100 and CINIC10 Datasets*

| Method | CIFAR10 | | CIFAR100 | | CINIC10 | |
|---|---|---|---|---|---|---|
| | Average Accuracy | Parameter Size | Average Accuracy | Parameter Size | Average Accuracy | Parameter Size |
| Local Adaptation - ResNet18 | 0.87±0.02 | 11M | 0.61±0.03 | 11M | 0.61± 0.05 | 11M |
| Ditto - ResNet18 | 0.90±0.03 | 11M | 0.57±0.03 | 11M | 0.64 ± 0.05 | 11M |
| perFedAvg - ResNet18 | 0.91±0.02 | 11M | 0.64±0.03 | 11M | 0.59 ± 0.03 | 11M |
| Local Adaptation - DARTS | 0.85 ±0.03 | 4.7M | 0.63 ±0.04 | 4.7M | 0.73 ±0.04 | 4.7M |
| Ditto - DARTS | 0.77±0.05 | 4.7M | 0.45±0.04 | 4.7M | 0.60±0.03 | 4.7M |
| SPIDER | **0.93±0.02** | **2.8M** | **0.68±0.03** | **3.1M** | **0.84±0.07** | **3.3M** |

**Implementation and Deployment**   We implement the proposed method for distributed computing with nine nodes, each equipped with GPUs. We set this as a cross-silo FL setting with one node representing the server and eight nodes representing the clients. These client nodes can represent real-world organizations such as hospitals and clinics that aim to collaboratively search for personalized architectures for local benefits such as higher accuracy in a privacy-preserving FL manner. Since we are working on a cross-silo setting, neural architecture search cost may not be a concern since devices in cross-silo are rich in resources.

## 5.2 RESULTS

Here, we report the comparison of our proposed method SPIDER with the other state-of-the-art personalized methods; Ditto, perFedAvg, and local adaptation. Since these schemes use a pre-defined architecture, we use the Reset18 model because of its comparable model size. Since we are exploiting DARTS based search space in this work, we also use a DARTS model Liu et al. (2018) searched in a centralized setting on CIFAR10 dataset as our base model for local adaptation and Ditto personalization schemes.

### 5.2.1 AVERAGE TEST ACCURACY

As illustrated in Table 1, our method, SPIDER, achieves the objective **R1** by outperforming the state-of-the-art personalization methods; Ditto, local adaptation and perFedAvg on three image classification datasets CIFAR10, CIFAR100 and CINIC10. For CIFAR100 and CINIC10 dataset, Local adaptation with the DARTS based model performed the second best and outperformed Ditto, perFedAvg and Local adaption on ResNet18. For CIFAR10, perFedAvg with Resnet18 performed the second best. Overall, we obtain 2%, 4% and 11% higher accuracy as compared to the best performing scheme from the representative state-of-the-art personalization algorithms; Ditto, local adaptation, and perFedAvg, on the CIFAR10, CIFAR100 and CINIC-10 datasets, respectively. For CINIC10 dataset, we observe substantial performance gain (11%) with respect to the sota personalization techniques. We attribute this gain to the potential of SPIDER to adapt to invisible data better

by tailoring architectures for each client according to their data distribution. In addition, Ditto with DARTS model did not perform well for all three datasets and may need a bigger hyper-parameter search set than the one we considered (given in Appendix A.2) to converge better.

For personalization, in addition to average accuracy, the standard deviation (std) is also considered an important metric. Therefore, we also report the standard deviation for each method in Table 1. We note that the standard deviation across clients is almost similar for CIFAR10 and CIFAR100 datasets. For CINIC10, the standard deviation of SPIDER is slightly higher but this is achieved at 11% higher accuracy. Overall, we achieve almost the same standard deviation as other baselines but with the benefit of providing higher average test accuracy.

### 5.2.2 ARCHITECTURE HETEROGENEITY AND PERSONALIZATION GAIN

An important feature of SPIDER is that it helps each client search for its own architecture tailored for its own specific data distribution. Due to data heterogeneity across clients, we observe architectures to be heterogeneous across clients as shown in Appendix B.4, hence we achieve the objective **R2**, the search and training of heterogeneous architectures across clients. The objective **R3**, no dependence on auxiliary data, is obtained by the algorithmic framework of SPIDER as it does not rely on any auxiliary data and exploits only the client's dataset for searching and training. A byproduct of using DARTS search space is that the average parameter size of the models obtained with SPIDER is quite smaller. In order to further investigate whether the architecture searched by one client is best suited for its own data distribution, we perform the following experiment.

For any given client $i$, we apply its final architecture with its learned weights to another client $j$'s data and finetune $i$'s architecture on $j$'s data. We denote the accuracy we obtain on data $j$ via architecture $i$ as $AP_{ij}$. We calculate the architecture $i$ personalization gain or drop (if negative) as follows,

$$\boldsymbol{g}_i = \frac{\sum_{j=0, j \neq i}^{c-1}(AP_{ii} - AP_{ji})}{c-1} \tag{7}$$

We calculate it across all silos. The quantity $\boldsymbol{g}_i$ signifies the personalization gain of architecture $i$ across other silos architectures on its own dataset. Next, we calculate the personalization gain of SPIDER scheme as the mean of all $\boldsymbol{g}_i$, i.e., $\frac{\sum_{i=0}^{c-1} \boldsymbol{g}_i}{c}$, where $c$ is the total number of clients in the network. We conduct this experiment on CIFAR10, CIFAR100, CINIC10 datasets, where we finetune the architecture learned via SPIDER on other clients' data for 20 epochs and report the best accuracy. We obtain 1.96%, 3.11%, 3.71% personalization gain on CIFAR10, CIFAR100 and CINIC10 datasets, respectively. We observe that for the majority of the clients, its searched architecture outperforms the other architectures. This highlights the importance of architecture personalization which can be more powerful than just weight personalization (as shown by our empirical results). We report the $AP_{ij}$ and $AP_i$ values for all $i$ and $j$ in Table 2 and Appendix section B.4.

*Table 2: CINIC10: Personalization Gain Analysis*

*(a) Accuracy values of Architecture i on j's data (where i and j represent client IDs.)*

| (j, i) | i = 0 | i = 1 | i = 2 | i = 3 | i = 4 | i = 5 | i = 6 | i = 7 |
|---|---|---|---|---|---|---|---|---|
| j = 0 | 90.59 | 88.98 | 88.65 | 88.75 | 89.56 | 89.17 | 89.17 | 88.27 |
| j = 1 | 86.54 | 90.55 | 87.42 | 87.79 | 87.59 | 87.94 | 87.21 | 86.25 |
| j = 2 | 72.79 | 73.27 | 78.23 | 73.49 | 75.21 | 74.62 | 74.73 | 73.81 |
| j = 3 | 83.71 | 84.27 | 84.22 | 88.35 | 84.68 | 84.82 | 84.63 | 84.43 |
| j = 4 | 69.46 | 71.40 | 70.34 | 71.69 | 75.29 | 70.58 | 71.52 | 67.92 |
| j = 5 | 73.90 | 71.95 | 74.45 | 73.16 | 74.02 | 77.53 | 74.18 | 72.03 |
| j = 6 | 83.21 | 83.69 | 84.29 | 82.40 | 83.78 | 82.92 | 88.48 | 81.76 |
| j = 7 | 89.5 | 88.68 | 89.81 | 88.37 | 89.5 | 89.19 | 88.44 | 91.25 |

*(b) Accuracy Gain/Drop Matrix ($AP_{jj} - AP_{ij}$) and the resultant $\boldsymbol{g}_j$ vector*

| (j, i) | i = 0 | i = 1 | i = 2 | i = 3 | i = 4 | i = 5 | i = 6 | i = 7 | $\boldsymbol{g}_j$ |
|---|---|---|---|---|---|---|---|---|---|
| j = 0 | 0. | 1.61 | 1.94 | 1.84 | 1.03 | 1.42 | 1.42 | 2.32 | 1.65 |
| j = 1 | 4.01 | 0. | 3.13 | 2.76 | 2.96 | 2.61 | 3.34 | 4.3 | 3.30 |
| j = 2 | 5.44 | 4.96 | 0. | 4.74 | 3.02 | 3.61 | 3.5 | 4.42 | 4.24 |
| j = 3 | 4.64 | 4.08 | 4.13 | 0. | 3.67 | 3.53 | 3.72 | 3.92 | 3.96 |
| j = 4 | 5.83 | 3.89 | 4.95 | 3.6 | 0. | 4.71 | 3.77 | 7.37 | 4.87 |
| j = 5 | 3.63 | 5.58 | 3.08 | 4.37 | 3.51 | 0. | 3.35 | 5.5 | 4.15 |
| j = 6 | 5.27 | 4.79 | 4.19 | 6.08 | 4.7 | 5.56 | 0. | 6.72 | 5.33 |
| j = 7 | 1.75 | 2.57 | 1.44 | 2.88 | 1.75 | 2.06 | 2.81 | 0. | 2.18 |

## 6 CONCLUSION

We proposed SPIDER, an algorithmic framework that can search personalized neural architecture for FL. SPIDER specializes a weight-sharing-based global regularization to perform progressive neural architecture search. Experimental results demonstrate that SPIDER outperforms other state-of-the-art personalization methods by searching and training a personalized architecture for each client.

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
