# OpenReview forum: "SPIDER: Searching Personalized Neural Architecture for Federated Learning"
_ICLR.cc/2023/Conference — Submitted to ICLR 2023_

### Official Review · Reviewer_fLM4 · 2022-10-25

**Confidence:** 3
**Correctness:** 3
**Technical Novelty And Significance:** 2
**Empirical Novelty And Significance:** 2
**Recommendation:** 3

**Clarity, Quality, Novelty And Reproducibility:**

There are some rooms for improving the quality, clarity, and originality of the proposed method. For example, detailed comparison with some existing methods that are utilizing the supernet for Federated NAS (i.e. HeteroFL, FedorAS, etc) and more analysis for the proposed method (i.e. ablation study)

**Strength And Weaknesses:**

**Strength**
- The paper is easy to read.
- They tackle the practical federated learning with architectural personalization.
- The results are impressive.

**Weaknesses**
- The proposed algorithm trains two models at each client, both global supernet and local model, which may increase the computation and memory consumption of local resources. Although NAS is devised to optimize consumption of given resources, the proposed framework seems to be inefficient and even not practical in real-world scenarios.
- In the same context, the overall communication and computation costs should be discussed and compared with the baseline models. But, I can only find the comparison of the searched model’s performance and its size (parameters).
- It is better to compare with more similar existing baseline models that uses supernet for sampling the local models, i.e. HeteroFL [Diao et al 21] or FedorAS [Dudziak et al 22], in terms of both efficiency (communication costs, etc) and performance.
- More experiments should be conducted for detailed analysis, i.e. ablation study.

Diao et al, HeteroFL: Computation and Communication Efficient Federated Learning for Heterogeneous Clients, ICLR 2021.

Dudziak et al, FedorAS: Federated Architecture Search under system heterogeneity


**Summary Of The Paper:**

The authors tackle personalized neural architecture search for federated learning, where data heterogeneity is alleviated by the architectural personalization. The authors propose a method named SPIDER that uses a supernet from which local models can be sampled while sharing the weights. The authors validate their method on several benchmarks against existing methods.


**Summary Of The Review:**

I enjoyed reading the paper, but several improvements seem to be required, as mentioned above.

---

> ### Author Response · Authors · 2022-11-18
> **Response to Reviewer fLM4**
>
> We thank reviewer fLM4 for their review and comments. We answer below all their comments. Based on these answers, we would like to politely ask the reviewer to re-evaluate the revised paper. We are open to further discussion with the reviewer regarding the revised paper.
>
> ### 1. The proposed framework seems to be inefficient and even not practical in real-world scenarios.
>
> #### We propose SPIDER for a cross-silo setting such as hospitals that can afford high computing power and demand high accuracy. This work focuses on providing accuracy fairness for such cross-silo settings. We achieve this objective by personalizing the architecture at each silo. However, there is a tradeoff between accuracy and compute cost. As an example, with the proposed framework SPIDER, we can achieve high average accuracy (93%) across silos (on the CIFAR10 dataset) by paying the price of high compute (3.4 GB peak memory cost), whereas FedorAS consumes (1.9GB) peak memory cost only by paying the price of lower accuracy (91%). In our ablation study, we find that if each silo performs centralized NAS and local training, they can achieve 91% accuracy as well. However, SPIDER assists each silo to search for a personalized architecture tailored to its own data distribution while also learning from other silos. This architecture personalization yields a 2% performance gain on the CIFAR10 dataset.
>
> ### 2. In the same context, the overall communication and computation costs should be discussed and compared with the baseline models. But, I can only find the comparison of the searched model’s performance and its size (parameters).
>
> #### We thank the reviewer for the insightful comment. We have added a discussion in Appendix section C to compare the training time and peak memory cost of various PFL schemes.
>
> ### 3. It is better to compare with more similar existing baseline models that uses supernet for sampling the local models, i.e. HeteroFL [Diao et al 21] or FedorAS [Dudziak et al 22], in terms of both efficiency (communication costs, etc) and performance.
>
> #### The proposed framework SPIDER is for a cross-silo setting, whereas HeteroFL and FedorAS are proposed for cross-device settings, and focus on clusters/tiers with different computing powers. These works assign architectures either manually (HetreoFL) or via NAS (FedorAS) based on the computing power of the edge device. In these works, silos belonging to the same clusters get the same architecture even if their data distribution is different. Contrary to these schemes, we focus on architecture assignments based on data distribution. We have added FedorAS [1] baseline in section C [Appedix], and we found that we achieve a 2% higher personalization gain as compared to FedorAS at the cost of a higher compute cost.
> #### [1] Dudziak et al, FedorAS: Federated Architecture Search under system heterogeneity.
> #### [2] Diao et al, HeteroFL: Computation and Communication Efficient Federated Learning for Heterogeneous Clients, ICLR 2021.
>
> ### 4. More experiments should be conducted for detailed analysis, i.e. ablation study.
>
> #### We thank the reviewer for the feedback. We have added a detailed section, Section B, to study different components of the proposed method. To perform ablation, we compare our method to a basic setting of NAS, which is a centralized NAS. Next, we compare it to Federated NAS, which searches a global architecture in a federated manner and then trains it via FedAvg. In the next experiment, we add an l2 regularization-based PFL scheme to train the global architecture searched via Federated NAS. This investigation helps us appreciate the benefits of architecture personalization in both centralized and federated learning settings. This study also highlights that a global neural architecture search may not be enough even if combined with a PFL scheme for training. To further emphasize it, we also perform architecture personalization experiments on the architectures searched and trained via SPIDER and found that architecture personalization can provide a gain of 3.11% and 3.71% to each silo (on average) for CIFAR100 and CINIC10 datasets, respectively.

---

### Official Review · Reviewer_pg98 · 2022-10-25

**Confidence:** 5
**Clarity, Quality, Novelty And Reproducibility:** The quality, clarity and originality …
**Correctness:** 4
**Technical Novelty And Significance:** 2
**Empirical Novelty And Significance:** 3
**Recommendation:** 5

**Strength And Weaknesses:**

S1. The topic studied in this paper, i.e., realizing both weight and architectural personalization in the federated learing (FL) scenarios, is important and novel in the field of FL.
S2. This paper is well-written, the problem definition and algorithm description are clear and easy to follow.
S3. In the experiments, the authors compare the proposed method with the personalized FL methods with a fixed model architecture, demonstrating the significance of realizing architectural personalization in the field of FL.

W1. A kind of solution for the problem studied in this paper is to combine the existing personalized federated training method with the existing gradient-based neural architecture search (NAS). That is using the personalized federated training method to replace the non-personalized single-machine model training method applied in the existing gradient-based NAS method. In this way, both the architectures and the weights can be optimized in a personalized manner under the FL framework, realizing both weight and architectural personalization in the FL scenarios.
Authors should compare the above solution (using different NAS algorithms and federated training methods) with the proposed method in the experimental part, to demonstrate the superiority and the importance of the proposed method.
W2. What’s the difference between the solution mentioned in W1 with the proposed method in this paper? The solution mentioned in W1 is straightforward (it is just a simple combination of the existing works), and more appropriate solution should be designed to deal with the novel federated NAS problem. Does the novel federated NAS problem bring new challenges, requiring novel NAS solutions or federated training methods? Does the proposed method solve these challenges? Authors can discuss these contents in depth.
W3. The experimental analysis is not enough. The performance curves, i.e., the federated performance of the proposed method and other federated NAS methods at different time steps, are not given and not analyzed in the experimental parts. These curves can reflect the search effect and the search efficiency of different algorithms under the FL framework, which is important for evaluating a federated NAS algorithm.
W4. Why perturbation-based NAS is selected to realizing the NAS operations in the proposed method? Why not select the other gradient-based methods? The selection basis should be clarified in the next version. In addition, authors should add experiments to examine the differences of selecting perturbation-based NAS instead of other NAS method.

**Summary Of The Paper:**

This paper formulates a personalized neural architecture search framework for the federated learning, aiming at searching for the personalized neural architectures for each client under the federated learning scenarios. The topic and the proposed idea of this paper are novel, but the experimental analysis and the technical depth of this paper are not enough.

**Summary Of The Review:**

The topic is novel and important, the paper is well-written and the experiments demonstrate the significance of the proposed method. But some important baselines are missing in the experiments, the novelty of the proposed method should be discussed in depth, the experimental analysis are not enough, and the design of the proposed method are not clear. Suggest to further modify this paper for publication.

---

> ### Author Response · Authors · 2022-11-18
> **Response to Reviewer pg98**
>
> We thank reviewer pg98 for their review and comments. We answer below all their comments. Based on these answers, we would like to politely ask the reviewer to re-evaluate the revised paper. We are open to further discussion with the reviewer regarding the revised paper.
>
> ### W1. A kind of solution for the problem studied in this paper is to combine the existing personalized federated training method with the existing gradient-based neural architecture search (NAS).
>
> #### We thank the reviewer for the insightful comment. To address this comment, we have investigated two settings in our ablation study (Section B, Appendix). We evaluate federated neural architecture search where all silos search for a global architecture via federated learning and then train it in a federated manner via FedAvg. We keep the Perturbation-based NAS baseline the same for these two settings to make it a fair comparison. We explain our motivation for using Perturbation-based NAS in our answer to the W3 comment and subsection 4.4 in the main text). For this first setting, we achieve 86% average accuracy on the Federated NAS scheme. Next, we use the global model architecture found in the federated neural architecture search and train it via an l2 regularization-based PFL framework, Ditto, and achieve 87% accuracy. SPIDER outperforms these two settings that search a global architecture via federated learning and perform training in a non-personalized [setting 1] or personalized fashion [setting 2]. This shows that model weight parameters-based personalization alone may not be enough to achieve personalization in data heterogeneous settings. To investigate it further, we analyze the architecture personalization gains of SPIDER by deploying one silo architecture on other silos data, and finetuning (i.e, personalizing) it and found that each silo gains around 1.96%, 3.11%, and 3.71% performance gain on average if they use the personalized architectures searched via SPIDER  for CIFAR10, CIFAR100, and CINIC10 datasets, respectively.
>
> ### W2: What’s the difference between the solution mentioned in W1 with the proposed method in this paper
>
> #### In our ablation study of personalization gain, we found that architecture personalization can perform better than the combination of NAS with personalized FL such as Ditto. However, architecture personalization is not straightforward as weight personalization. Architecture personalization can yield different architectures due to data heterogeneity across different silos (as shown in Figure 6, Appendix) and the current PFL schemes may not work for this heterogeneous architecture search and train setting. This requires a framework for heterogeneous architecture aggregation that can personalize full architectures as well as model weights. As we found in our work, in the non-IID context, there can be no or minimal overlap between the architectures across silos. This can lead to inefficient aggregation. To address this issue, we exploit the connection overlap between the supernet and child architecture and use l2 regularization between these models for local model training. This setup not only personalizes architecture and model weights but also keeps them private in such a way that which architecture was selected at a particular silo remains unknown on the server side.
>
> ### W3: The experimental analysis is not enough. The performance curves, i.e., the federated performance of the proposed method and other federated NAS methods at different time steps, are not given and not analyzed in the experimental parts.
>
> #### We provide the performance curves of all the methods in Figure 2, Appendix. Figure 6b (Appendix) also provides insights about the architecture search and train phases. We have added a perturbation-based FedNAS baseline as well in our ablation study. In addition, we have also added a computational cost analysis in the units of training time and peak memory cost.

---

> > ### Author Response · Authors · 2022-11-18
> > **Response to Reviewer pg98**
> >
> > ### W4: Why perturbation-based NAS is selected to realizing the NAS operations in the proposed method? Why not select the other gradient-based methods?
> >
> > #### The gradient-based NAS methods are mostly based on bi-level optimization (optimization of architecture parameters and model parameters) [1]. Perturbation-based NAS [2] is a simple and efficient NAS method as it does not require bi-level optimization, and exploits validation accuracy for architecture search. This work has shown the magnitude of architecture parameters may not be a good parameter for operation selection. They have shown via extensive empirical analysis that perturbation-based NAS can outperform the previous state-of-the-art methods. It is for the simplicity and superior (state-of-the-art) performance of this algorithm that we select this method. More explanation can be found in Section 4.4 in our revised version.
> >
> > ##### [1] Liu, Hanxiao, Karen Simonyan, and Yiming Yang. "Darts: Differentiable architecture search." arXiv preprint arXiv:1806.09055 (2018).
> > ##### [2]. Wang, Ruochen, et al. "Rethinking architecture selection in differentiable NAS." Outstanding Paper Award, ICLR, (2021).

---

### Official Review · Reviewer_AAmq · 2022-10-26

**Confidence:** 5
**Correctness:** 3
**Technical Novelty And Significance:** 2
**Empirical Novelty And Significance:** 2
**Recommendation:** 3

**Clarity, Quality, Novelty And Reproducibility:**

the notations used are not well-presented, making the draft hard to follow. The notations used are not formal. Typically, matrix, vectors, and scalars are using different fonts for easy reading. However, all the notations are using the same fonts.

There are too many typos and grammar errors indicating the low quality of the draft. Even in the abstract, there are clear typos, e.g., “all the clientsin FL”.
More examples, in the intro “we demonstrate accuracy gain of 10%, 6%, 4% over”==>”we demonstrate an accuracy gain of 10%, 6%, 4% over “
In the related work “ meeting clients efficiency budgets”==> meeting clients’ efficiency budgets

There are no code release thus hard to validate the reproducibility.

**Strength And Weaknesses:**


Strong points:
S1: the results seem promising.

S2: the literature review is somehow thorough.

Weakness:

W1: the idea is not novel and the solution is straightforward. The submission claims in the related work section “Where all these models search for a unified global model, a key distinction of our work with these works is that we aim to search for a personalized model for each client.” However, this is not true. Many existing work such as the following two are also providing NAS for different clients in FL.

1). Personalized Neural Architecture Search for Federated Learning, Minh Hang et al. Neurips Workshop 2021.
2). Personalized Federated Learning via Heterogeneous Modular Networks. Tianchun Wang, et al. ICDM 2022, 2022.

Actually, these two papers are using more elegant solution for PFL. The proposed solution adopts the mixture method of ideas of NAS and MAML. It needs to maintain two networks (one global and one local) that causes additional communication/storage burden in FL. The second term (regularization) term in formula (5) is very straightforward.


W2: the notations used are not well-presented, making the draft hard to follow. The notations used are not formal. Typically, matrix, vectors, and scalars are using different fonts for easy reading. However, all the notations are using the same fonts.

W3: There are too many typos and grammar errors indicating the low-quality of the draft. Even in the abstract, there are clear typos, e.g., “all the clientsin FL”.
More examples, in the intro “we demonstrate accuracy gain of 10%, 6%, 4% over”==>”we demonstrate an accuracy gain of 10%, 6%, 4% over “
In the related work “ meeting clients efficiency budgets”==> meeting clients’ efficiency budgets

W4: The experiments are far from complete. The authors only compared to very few PFL baselines. Many more related works are not compared. For example, the most relevant “Personalized Neural Architecture Search for Federated Learning, Minh Hang et al. Neurips Workshop 2021.” Was not compared.

W5: There is no theoretical analysis if the algorithm will converge or not.


**Summary Of The Paper:**

The submission proposed NAS algorithm for personalized Federated Learning. Specifically, it alternately optimizing one homogeneous global model and one local model that is the subnetwork of the global one. Basically, the idea is not novel and the solution is straightforward, limiting its contribution to the community. Moreover, the notations used are not well-presented, making the draft hard to follow. I thus do not champion the acceptance.

**Summary Of The Review:**

The submission proposed NAS algorithm for personalized Federated Learning. Specifically, it alternately optimizes one homogeneous global model and one local model which is the subnetwork of the global one. Basically, the idea is not novel and the solution is straightforward, limiting its contribution to the community. Moreover, the notations used are not well-presented, making the draft hard to follow. The experiments are also very short. Many baselines are not compared. I thus do not champion acceptance.

---

> ### Author Response · Authors · 2022-11-18
> **Response to Reviewer AAmq**
>
> We thank reviewer AAmq for their review and comments. We answer below all their comments. Based on these answers, we would like to politely ask the reviewer to re-evaluate the revised paper. We are open to further discussion with the reviewer regarding the revised paper.
>
> ### W1: Comparison to existing works
>
> #### The first work [1] of PFL keeps a personalized and global component of the architecture. To draw a boundary between these two components is a heuristic search and avoiding this search has been one of our motivations for the proposed approach (as stated in Objective R2, Section3). In addition, work [1] focuses only on IID Computer vision tasks and has not explored the non-IID data distribution setting that is the realistic setting for PFL works. The other work [2] is a very recent work (published on arXiv after our ICLR submission).
> #### From the novelty perspective, we believe our work is novel as the proposed SPIDER-Searcher and SPIDER-Trainer allow the search and training of heterogeneous and fully personalized architectures in a federated manner. It outperforms the representative state-of-the-art personalized federated learning methods by an accuracy margin of 2%, 5%, and 11% on the CIFAR10, CIFAR100, and CINIC10 datasets, respectively. This shows the significance of personalization at the architectural level as compared to personalization only at the model-weight level. By the design of the SPIDER framework, the personalized architecture as well as its weight parameters remain unknown on the server side as they are never shared with the server during training.  From our ablation studies, we further investigate the architectures we find via SPIDER by finetuning one silos architecture on the other silos data. We find that we achieve 1.96%, 3.11%, and 3.71% architecture personalization gain per silo (on average) for the CIFAR10, CIFAR100, and CINIC10 datasets, respectively. This highlights that the personalized architectures searched via SPIDER are tailored for silos data distribution.
> #### [1]. Personalized Neural Architecture Search for Federated Learning, Minh Hang et al. Neurips Workshop 2021.
> #### [2]. Personalized Federated Learning via Heterogeneous Modular Networks. Tianchun Wang, et al. ICDM 2022.
>
>
> ### W2: the notations used are not well-presented, making the draft hard to follow.
>
> #### We thank the reviewer for his/her valuable comment. We have revised our manuscript to address this issue. We have used capital letters without bold notation for Matrix, small letters with bold notations for vectors, and small letters without bold notations for scalars in our revised manuscript.
>
>
> ### W3: There are too many typos and grammar errors indicating the low-quality of the draft.
>
> #### We thank the reviewer for highlighting the typos and errors. We have revised our manuscript for these mistakes.
>
> ### W4: Comparison to  Personalized Neural Architecture Search for Federated Learning Hang et al. Neurips Workshop 2021 Work
>
> #### We did not add [1] as this work only considered IID vision tasks in their experiment settings and our focus is on non-IID data settings. However, we have added two more baselines FedorAS [2] and FedNAS [3] in our revised version.
>
> #### [1]. Personalized Neural Architecture Search for Federated Learning Hang et al. Neurips Workshop 2021.
> #### [2]. Dudziak, Lukasz, Stefanos Laskaridis, and Javier Fernandez-Marques. "FedorAS: Federated Architecture Search under system heterogeneity." arXiv preprint arXiv:2206.11239 (2022).
> #### [3] He, Chaoyang, Murali Annavaram, and Salman Avestimehr. "Towards non-iid and invisible data with fednas: federated deep learning via neural architecture search." arXiv preprint arXiv:2004.08546 (2020).
>
> ### W5: There is no theoretical analysis if the algorithm will converge or not.
>
> #### We have provided comprehensive experimental results on three datasets, along with the convergence curves in Sections A and B (Appendix). We aim to study the theoretical analysis of our work in future work.

---

> > ### Comment · Reviewer_AAmq · 2022-11-21
> > **Novelty concerns**
> >
> > The rebuttal was revised according to my comments. However, the important novelty concerns are not yet addressed and discussed. I thus keep my original score.

---

> > > ### Author Response · Authors · 2022-12-07
> > > **Response to Novelty concerns**
> > >
> > > Dear Reviewer, thank you for reviewing our revised version. We discussed the novelty concerns in our response above to your first comment where we combined comparison to existing works reviewer mentioned and our proposed method's novelty aspect. Here, we provide a more detailed response to the novelty aspect and we would like to politely ask the reviewer to re-evaluate the revised paper.
> > >
> > > >From the novelty perspective, we believe our work is novel as it proposes a different and complimentary approach, Search Personalized neural architecture for federated learning, to address the data heterogeneity challenge in FL. We propose SPIDER-Searcher to search fully^ personalized aarchitectures tailored for each clients data distribution and SPIDER-Trainer that trains the searched heterogeneous architectures in a federated manner. We show from empirical results on three datasets that SPIDER outperforms the representative state-of-the-art personalized federated learning methods by an accuracy margin of 2%, 5%, and 11% on the CIFAR10, CIFAR100, and CINIC10 datasets, respectively. This shows the significance of personalization at the architectural level as compared to personalization only at the model-weight level.
> > >
> > > >We also perform architecture personalization analysis study on three datasets where for any given client i, we apply its final architecture with its learned weights to another client j’s data and finetune i’s architecture on j’s data. We find that with SPIDER, we achieve 1.96%, 3.11%, and 3.71% architecture personalization gain per silo (on average) for the CIFAR10, CIFAR100, and CINIC10 datasets, respectively. This highlights that the personalized architectures searched via SPIDER are tailored for each silos data distribution.
> > >
> > > >In an ablation study, we also compare our work to FedNAS [1] and FedNAS combined with personalization scheme, Ditto and show that SPIDER can outperform a global architecture search and training even when combined with model-weight parameters based personalization. This also highlights that architecture personalization is important to address data heterogeneity issue in FL.
> > >
> > > >[1]. He, Chaoyang, et al. "Fednas: Federated deep learning via neural architecture search." (2021).
> > >
> > > >[2]. Li, Tian, et al. "Ditto: Fair and robust federated learning through personalization." International Conference on Machine Learning. PMLR, 2021.
> > >
> > > >^(without any split of local and global parameters of an architecture).

---

### Official Review · Reviewer_p6Ea · 2022-10-28

**Confidence:** 4
**Correctness:** 2
**Technical Novelty And Significance:** 2
**Empirical Novelty And Significance:** 2
**Recommendation:** 3

**Clarity, Quality, Novelty And Reproducibility:**

Clarity: this paper is mostly clear, though several parts require some clarifications. There are also some typos and formatting problems.

Quality: the paper can still be further improved. Some more discussions are needed to make the proposed method complete, and the experiments also need some improvements to fully evaluate the proposed method.

Novelty: the proposed method is novel and substantially different from existing methods in my opinion.

Reproducibility: this paper provides some information on its hyper-parameters, but does not provide its code, which may still create some difficulties repeating its experiments.


**Strength And Weaknesses:**

Strength:
- This paper proposes a novel method for federated neural architecture search
- Empirical results against several existing baseline methods show some improvements of the proposed method

Weakness:
- There are some unclear parts on the proposed method:
  - I am a bit confused on the definition of $a_k$, i.e. the architectural parameter for each client. Is this 0/1 variables whose elements are associated with operations in supernet? The authors may need to clarify that and revise some notations.
  - Based on the above concern, how is $a_k$ updated in Algorithm 2? Specifically, in step 7, the authors evaluate the validation accuracy of $a_k^t$ with one operation removed. Nevertheless, note that only one operation is kept in each iteration, then how can we evaluate the validation accuracy of $a_k^t$ with more operation removed? This is very confusing and require some clarifications.
- Some more discussions are needed for the proposed method:
  - I am a bit uncertain on the differences between SPIDER and the following method: perform supernet training, then distill useful subnetworks from the trained supernet based on local data, while SPIDER distills subnetworks during training. The authors may add some discussions on this variant.
  - Moreover, can algorithm 2 uses other NAS methods (e.g., [1])? Some discussions may also be needed to make the proposed method more complete.
- Some recent baseline methods [2, 3] are missing in current version, which makes it hard to evaluate the actual performance of the proposed method. The authors may need to add some comparisons to them
- Ablation studies are also largely missing in current version. It is unclear how each part of the proposed method contributes to its empirical performances.
- Minor issue:
  - It may be better to move some parts of the supplementary material to main text, e.g., architecture personalization gain, which seems critical for the whole paper
  - The authors may check the citation format. Currently citations are mixed with the main text and creates some difficulties in reading
  - Typos in abstract: line 6 “clientsin -> clients in”?

[1] Single Path One-Shot Neural Architecture Search with Uniform Sampling. ECCV 2020

[2] Personalized Federated Learning through Local Memorization. ICML 2022

[3] Federated Learning with Partial Model Personalization. ICML 2022


**Summary Of The Paper:**

This paper proposes SPIDER, an algorithmic framework that aims to search personalized neural architecture for federated learning. SPIDER alternatively optimizes a global model (called supernet) and a local model that is connected to global model and has its own architecture. Experimental results demonstrate that SPIDER outperforms other state-of-the-art personalization methods with much fewer times of hyperparameter tuning.

**Summary Of The Review:**

While this paper proposes a novel method and empirical results show some improvements, the proposed method needs more clarifications and discussions. For the empirical results, some recent methods are missing in comparison, and ablation studies are also largely missing. Thus I choose reject as my initial score.

---

> ### Author Response · Authors · 2022-11-18
> **Response to Reviewer p6Ea**
>
> We thank reviewer p6Ea for their review and comments. We answer below all their comments. Based on these answers, we would like to politely ask the reviewer to re-evaluate the revised paper. We are open to further discussion with the reviewer regarding the revised paper.
>
> ### 1. There are some unclear parts on the proposed method:
>
> #### We have revised the draft to add clarifications on the architecture search. In the revised draft, we denote the supernet with $\mathcal{A}$ and the child architecture with $\mathcal{A}_k$ notation. Child architecture gets updated following the mask, $\mathcal{A}_k = M_k \odot\mathcal{A}$.
> #### In the mask $M_k$, each entry is represented as $\alpha_{ij}$ which is the value associated with edge $i$ and operation $j$. For example, at a particular edge $i$ we have two operations to evaluate. That is, we have two $\alpha$ values to evaluate $\alpha_{i0}$ and $\alpha_{i1}$. First, we evaluate the validation accuracy of the child architecture by removing $\alpha_{i0}$, i.e, by making its value 0. Next, we make  $\alpha_{i0}$ 1 again and remove $\alpha_{i1}$ (, i.e, by making its value 0). The operation whose removal gives the highest accuracy drop, the highest impact, on the child network is selected. The rest of the operations on that particular edge are removed in the child network $\mathcal{A}_k$. More details can be found in subsection 4.2 in the main text and A.3 in the Appendix.
>
>
> ### 2. Some more discussions are needed for the proposed method
>
> #### We thank the reviewer for the feedback.
> ##### One challenge for the suggested ‘distilled supernetwork towards the end approach is that after the distillation of subnetworks, these networks need to be trained further. For training in a federated setting, conventional methods such as FedAvg aggregation might not work since architectures can be heterogeneous across silos due to data heterogeneity and can have no to minimal overlap between them. Therefore, we propose to distill the network in the initial 400 rounds and train the personalized subnetworks for the next 500 rounds. Hence, our method provides a sophisticated framework that allows both the search and training of heterogeneous architectures in a federated manner. It enables the personalization of architectures in such a way that they are not shared with the server.
>
>
> ##### Our understanding is that any gradient-based NAS could work for (step 11) in the proposed Algorithm [1] of the revised manuscript to update the child's local architecture. However, our goal was to select a simple but effective method. For example, the suggested NAS method  [1] can work where instead of gradual pruning of operation, all edges are pruned in one step to obtain the child network. However, contrary to this, we allow the child network to recover (for about 20 rounds) after each pruning step for better operation selection in the next pruning step. This brings stability to the architecture search. Overall, we select the Perturbation-based NAS method [2] that performs gradual pruning for two main reasons. First, it is a simple NAS method because it does not require bi-level optimization that can increase the compute cost or training time. Second, this method is an efficient method as it does not require any alpha (architecture) parameter training and outperforms the alpha parameters magnitude-based architecture selection methods.
>
> ##### [1] Single Path One-Shot Neural Architecture Search with Uniform Sampling. ECCV 2020.
> ##### [2] Wang, Ruochen, et al. "Rethinking architecture selection in differentiable NAS." Outstanding Paper Award, ICLR, (2021).
>
>
> ### 3. Some recent baseline methods are missing in the current version.
>
> #### We have three baselines, local adaptation, perFedAvg, and Ditto, of PFL in our work. We have added FedNAS [1] and FedorAS [2] baselines for the CIFAR10 dataset in our revised version (Appendix Sections B and C). Local Memorization PFL baseline and partial model personalization can be added later.
>
> ###### [1] He, Chaoyang, Murali Annavaram, and Salman Avestimehr. "Towards non-iid and invisible data with fednas: federated deep learning via neural architecture search." arXiv preprint arXiv:2004.08546 (2020).
> ###### [2] Dudziak, Lukasz, Stefanos Laskaridis, and Javier Fernandez-Marques. "FedorAS: Federated Architecture Search under system heterogeneity." arXiv preprint arXiv:2206.11239 (2022).

---

> > ### Author Response · Authors · 2022-11-18
> > **Response to Reviewer p6Ea**
> >
> > ### 4. Ablation studies are also largely missing in current version.
> >
> > #### We thank the reviewer for his/her valuable suggestion. We have added a detailed section, Section B, to study different components of the proposed method. To perform ablation, we compare our method to a basic setting of NAS, which is a centralized NAS. Next, we compare it to Federated NAS, which searches a global architecture in a federated manner and then trains it via FedAvg. In the next experiment, we add an l2 regularization-based PFL scheme to train the global architecture searched via Federated NAS. This investigation helps us appreciate the benefits of architecture personalization in both centralized and federated learning settings. This study also highlights that a global neural architecture search may not be enough even if combined with a PFL scheme for training. To further emphasize it, we also perform architecture personalization experiments on the architectures searched and trained via SPIDER and found that architecture personalization can provide a gain of 3.11% and 3.71% to each silo (on average) for CIFAR100 and CINIC10 datasets, respectively.
> >
> > ### 5. Minor issue:
> >
> > #### We thank the reviewer for highlighting the significance of the architecture personalization gain experiment. We have moved it to the main text. We have corrected the typos in the revised manuscript as well. For the citation format, we follow the ICLR citation style.

---

### Author Response · Authors · 2022-11-18
**Revision Summary**

We thank all reviewers for their time and their feedback. Below we summarize the major revisions. We also provide detailed answers to each reviewer in separate comments. We submitted a revised version of the paper taking into account the reviewers’ remarks. All the major revisions are highlighted in blue color text. We would be happy to further exchange with reviewers during the discussion period.

Summary of the revisions:

#### 1. We have added ablation studies on the CIFAR10 dataset with the various settings of NAS as suggested by reviewer p6Ea, fLM4, pg98. The ablation study can be found in the Appendix Section B of the revised draft.
#### 2. To highlight the significance of architecture personalization, we have moved the Personalization Gain analysis on the CINIC10 dataset from the discussion to the main text Section 5.2.2 as recommended by Reviewer p6Ea. We have also added two tables for the personalization gain analysis on CIFAR100 and CINIC10 datasets in Appendix Section B.4.
#### 3. To discuss the tradeoff between performance and compute cost, we have added Computational Cost Analysis in Appendix Section C as suggested by Reviewer fLM4.
#### 4. We have added a NAS-based federated baseline, FedorAS, in computational cost analysis Section C as suggested by reviewer ffLM4 and pg98.
#### 5. We have also revised the notations in the methods section to add clarity as recommended by reviewer p6Ea, AAmq.

---

### Decision · Program_Chairs · 2023-01-20

**Decision:**

Reject

**Justification For Why Not Higher Score:**

1. While the problem can be interesting and important, there is little technical novelty in this paper. More discussion with related works should be included to draw a distinct boundary between existing works and the proposed SPIDER.
2. Presentation should be improved.

**Justification For Why Not Lower Score:**

N/A

**Metareview: Summary, Strengths And Weaknesses:**

This paper proposes a NAS algorithm called SPIDER that targets at finding personalized architectures for different client under federated learning (FL) setting. The challenges here are data heterogeneity and data invisibility. SPIDER alternately optimizes one homogeneous global supernet and one local model that is the subnetwork of the supernet. Specifically, a progressive search strategy is utilized for local clients. Experiments on several benchmark datasets show the promising performance of the proposed SPIDER.

Strengths
- The topic is important and novel.

Weaknesses
- Experimental analysis and the technical depth of this paper are not enough
- Presentation is hard to follow. Notations used are not well-presented